# Adversarial Prediction Games for Multivariate Losses

**Hong Wang**      **Wei Xing**      **Kaiser Asif**      **Brian D. Ziebart**
Department of Computer Science
University of Illinois at Chicago
Chicago, IL 60607
{hwang27, wxing3, kasif2, bziebart}@uic.edu

## Abstract

Multivariate loss functions are used to assess performance in many modern prediction tasks, including information retrieval and ranking applications. Convex approximations are typically optimized in their place to avoid NP-hard empirical risk minimization problems. We propose to approximate the training data instead of the loss function by posing multivariate prediction as an adversarial game between a loss-minimizing prediction player and a loss-maximizing evaluation player constrained to match specified properties of training data. This avoids the non-convexity of empirical risk minimization, but game sizes are exponential in the number of predicted variables. We overcome this intractability using the double oracle constraint generation method. We demonstrate the efficiency and predictive performance of our approach on tasks evaluated using the precision at k, the F-score and the discounted cumulative gain.

## 1   Introduction

For many problems in information retrieval and learning to rank, the performance of a predictor is evaluated based on the combination of predictions it makes for multiple variables. Examples include the precision when limited to $k$ positive predictions (P@k), the harmonic mean of precision and recall (F-score), and the discounted cumulative gain (DCG) for assessing ranking quality. These stand in contrast to measures like the accuracy and (log) likelihood, which are additive over independently predicted variables. Many multivariate performance measures are not concave functions of predictor parameters, so maximizing them over empirical training data (or, equivalently, empirical risk minimization over a corresponding non-convex multivariate loss function) is computationally intractable [11] and can only be accomplished approximately using local optimization methods [10]. Instead, convex surrogates for the empirical risk are optimized using either an additive [21, 12, 22] or a multivariate approximation [14, 24] of the loss function. For both types of approximations, the gap between the application performance measure and the surrogate loss measure can lead to substantial sub-optimality of the resulting predictions [4].

Rather than optimizing an approximation of the multivariate loss for available training data, we take an alternate approach [26, 9, 1] that robustly minimizes the exact multivariate loss function using approximations of the training data. We formalize this using a zero-sum game between a predictor player and an adversarial evaluator player. Learned weights parameterize this game's payoffs and enable generalization from training data to new predictive settings. The key computational challenge this approach poses is that the size of multivariate prediction games grows exponentially in the number of variables. We leverage constraint generation methods developed for solving large zero-sum games [20] and efficient methods for computing best responses [6] to tame this complexity. In many cases, the structure of the multivariate loss function enables the zero-sum game's Nash equilibrium to be efficiently computed. We formulate parameter estimation as a convex optimization problem and solve it using standard convex optimization methods. We demonstrate the benefits of this approach on prediction tasks with P@k, F-score and DCG multivariate evaluation measures.

## 2 Background and Related Work

### 2.1 Notation and multivariate performance functions

We consider the general task of making a multivariate prediction for variables $\mathbf{y} = \{y_1, y_2, \ldots, y_n\} \in \mathcal{Y}^n$ (with random variables denoted as $\mathbf{Y} = \{Y_1, Y_2, \ldots, Y_n\}$) given some contextual information $\mathbf{x} = \{x_1, x_2, \ldots, x_n\} \in \mathcal{X} = \{X_1, X_2, \ldots, X_n\}$ (with random variable, $\mathbf{X}$). Each $x_i$ is the information relevant to predicted variable $y_i$. We denote the estimator's predicted values as $\hat{\mathbf{y}} = \{\hat{y}_1, \hat{y}_2, \ldots, \hat{y}_n\}$. The multivariate performance measure when predicting $\hat{\mathbf{y}}$ when the true multivariate value is actually $\mathbf{y}$ is represented as a scoring function: $\mathrm{score}(\hat{\mathbf{y}}, \mathbf{y})$. Equivalently, a complementary loss function for any score function based on the maximal score can be defined as: $\mathrm{loss}(\hat{\mathbf{y}}, \mathbf{y}) = \max_{\mathbf{y}', \mathbf{y}''} \mathrm{score}(\mathbf{y}', \mathbf{y}'') - \mathrm{score}(\hat{\mathbf{y}}, \mathbf{y})$.

For information retrieval, a vector of retrieved items from the pool of $n$ items can be represented as $\hat{\mathbf{y}} \in \{0, 1\}^n$ and a vector of relevant items as $\mathbf{y} \in \{0, 1\}^n$ with $\mathbf{x} = \{x_1, x_2, \ldots, x_n\}$ denoting side contextual information (e.g., search terms and document contents). Precision and recall are important measures for information retrieval systems. However, maximizing either leads to degenerate solutions (predict all to maximize recall or predict none to maximize precision). The precision when limited to exactly $k$ positive predictions, $\mathrm{P@k}(\hat{\mathbf{y}}, \mathbf{y}) = \frac{\hat{\mathbf{y}} \cdot \mathbf{y}}{k}$ where $||\hat{\mathbf{y}}||_1 = k$, is one popular multivariate performance measure that avoids these extremes. Another is the F-score, which is the harmonic mean of the precision and recall often used in information retrieval tasks. Using this notation, the F-score for a set of items can be simply represented as: $F_1(\hat{\mathbf{y}}, \mathbf{y}) = \frac{2\hat{\mathbf{y}} \cdot \mathbf{y}}{||\hat{\mathbf{y}}||_1 + ||\mathbf{y}||_1}$ and $F_1(\mathbf{0}, \mathbf{0}) = 1$.

In other information retrieval tasks, a ranked list of retrieved items is desired. This can be represented as a permutation, $\sigma$, where $\sigma(i)$ denotes the $i^{\text{th}}$-ranked item (and $\sigma^{-1}(j)$ denotes the rank of the $j^{\text{th}}$ item). Evaluation measures that emphasize the top-ranked items are used, e.g., to produce search engine results attuned to actual usage. The discounted cumulative gain (DCG) measures the performance of item rankings with $k$ relevancy scores, $y_i \in \{0, \ldots, k-1\}$ as: $\mathrm{DCG}(\hat{\sigma}, \mathbf{y}) = \sum_{i=1}^{n} \frac{2^{y_{\hat{\sigma}(i)}} - 1}{\log_2(i+1)}$ or $\mathrm{DCG}'(\hat{\sigma}, \mathbf{y}) = y_{\hat{\sigma}(1)} + \sum_{i=2}^{n} \frac{y_{\hat{\sigma}(i)}}{\log_2 i}$.

### 2.2 Multivariate empirical risk minimization

Empirical risk minimization [28] is a common supervised learning approach that seeks a predictor $\hat{P}(\hat{\mathbf{y}}|\mathbf{x})$ (from, e.g., a set of predictors $\Gamma$) that minimizes the loss under the empirical distribution of training data, denoted $\tilde{P}(\mathbf{y}, \mathbf{x})$: $\min_{\hat{P}(\hat{\mathbf{y}}|\mathbf{x}) \in \Gamma} \mathbb{E}_{\tilde{P}(\mathbf{y}, \mathbf{x}) \hat{P}(\hat{\mathbf{y}}|\mathbf{x})}[\mathrm{loss}(\hat{\mathbf{Y}}, \mathbf{Y})]$. Multivariate losses are often not convex and finding the optimal solution is computationally intractable for expressive classes of predictors $\Gamma$ typically specified by some set of parameters $\theta$ (e.g., linear discriminant functions: $\hat{P}(\hat{y}|x) = 1$ if $\theta \cdot \Phi(x, \hat{y}) > \theta \cdot \Phi(x, y') \; \forall y' \neq \hat{y}$).

Given these difficulties, convex surrogates to the multivariate loss are instead employed that are additive over $\hat{y}_i$ and $y_i$ (i.e., $\mathrm{loss}(\hat{\mathbf{y}}, \mathbf{y}) = \sum_i \mathrm{loss}(\hat{y}_i, y_i)$). Employing the logarithmic loss, $\mathrm{loss}(\hat{y}_i, y_i) = -\log \hat{P}(\hat{Y}_i = y_i)$ yields the logistic regression model [9]. Using the hinge loss yields support vector machines [5]. Structured support vector machines [27] employ a convex approximation of the multivariate loss over a training dataset $\mathcal{D}$ using the hinge loss function:

$$\min_{\theta, \xi_i \geq 0} ||\theta||^2 + \alpha \sum_i \xi_i \text{ such that } \forall i, \mathbf{y}' \in \mathcal{Y}, \theta \cdot [\Phi(\mathbf{x}^{(i)}, \mathbf{y}^{(i)}) - \Phi(\mathbf{x}^{(i)}, \mathbf{y}')] \geq \Delta(\mathbf{y}', \mathbf{y}^{(i)}) - \xi_i.$$

In other words, linear parameters $\theta$ for feature functions $\Phi(\cdot, \cdot)$ are desired that make the example label $\mathbf{y}^{(i)}$ have a potential value $\theta \cdot \Phi(\mathbf{x}^{(i)}, \mathbf{y}^{(i)})$ that is better than all alternative labels $\mathbf{y}'$ by at least the multivariate loss between $\mathbf{y}'$ and $\mathbf{y}^{(i)}$, denoted $\Delta(\mathbf{y}', \mathbf{y}^{(i)})$. When this is not possible for a particular example, a hinge loss penalty $\xi_i$ is incurred that grows linearly with the difference in potentials. Parameter $\alpha$ controls a trade-off between obtaining a predictor with lower hinge loss or better discrimination between training examples (the margin). The size of set $\mathcal{Y}$ is often too large for explicit construction of the constraint set to be computationally tractable. Instead, constraint generation methods are employed to find a smaller set of active constraints. This can be viewed as either finding the most-violated constraint [27] or as a loss-augmented inference problem [25]. Our

approach employs similar constraint generation techniques—in the inference procedure rather than the parameter learning procedure—to improve its efficiency.

## 3 Multivariate Prediction Games

We formulate a minimax game for multivariate loss optimization, describe our approach for limiting the computational complexity of solving this game, and describe algorithms for estimating parameters of the game and making predictions using this framework.

### 3.1 Game formulation

Following a recent adversarial formulation for classification [1], we view multivariate prediction as a two-player game between player $\hat{\mathbf{Y}}$ making predictions and player $\check{\mathbf{Y}}$ determining the evaluation distribution. Player $\hat{\mathbf{Y}}$ first stochastically chooses a predictive distribution of variable assignments, $\hat{P}(\hat{\mathbf{y}}|\mathbf{x})$, to maximize a multivariate performance measure, then player $\check{\mathbf{Y}}$ stochastically chooses an evaluation distribution, $\check{P}(\check{\mathbf{y}}|\mathbf{x})$, that minimizes the performance measure. Further, player $\check{\mathbf{Y}}$ must choose the relevant items in a way that (approximately) matches in expectation with a set of statistics, $\Phi(\mathbf{x}, \mathbf{y})$, measured from labeled data. We denote this set as $\Xi$.

**Definition 1.** *The* **multivariate prediction game (MPG)** *for $n$ predicted variables is:*

$$\max_{\hat{P}(\hat{\mathbf{y}}|\mathbf{x})} \min_{\check{P}(\check{\mathbf{y}}|\mathbf{x}) \in \Xi} \mathbb{E}_{\tilde{P}(\mathbf{x})\hat{P}(\hat{\mathbf{y}}|\mathbf{x})\check{P}(\check{\mathbf{y}}|\mathbf{x})} \left[ score(\hat{\mathbf{Y}}, \check{\mathbf{Y}}) \right], \tag{1}$$

*where $\hat{P}(\hat{\mathbf{y}}|\mathbf{x})$ and $\check{P}(\check{\mathbf{y}}|\mathbf{x})$ are distributions over combinations of labels for the $n$ predicted variables and the set $\Xi$ corresponds to the constraint: $\mathbb{E}_{\tilde{P}(\mathbf{x})P(\check{\mathbf{y}}|\mathbf{x})} \left[ \Phi(\mathbf{X}, \check{\mathbf{Y}}) \right] = \mathbb{E}_{\tilde{P}(\mathbf{y},\mathbf{x})} \left[ \Phi(\mathbf{X}, \mathbf{Y}) \right].$*

Since the set $\Xi$ constrains the adversary's multivariate label distribution over the entire distribution of inputs $\tilde{P}(\mathbf{x})$, solving this game directly is impractical when the number of training examples is large. Instead, we employ the method of Lagrange multipliers in Theorem 1, which allows the set of games to be independently solved given Lagrange multipliers $\theta$.

**Theorem 1.** *The multivariate prediction game's value (Definition 1) can be equivalently obtained by solving a set of unconstrained maximin games parameterized by Lagrange multipliers $\theta$:*

$$\max_{\hat{P}(\hat{\mathbf{y}}|\mathbf{x})} \min_{\check{P}(\check{\mathbf{y}}|\mathbf{x}) \in \Xi} \mathbb{E}_{\tilde{P}(\mathbf{x})\hat{P}(\hat{\mathbf{y}}|\mathbf{x})\check{P}(\check{\mathbf{y}}|\mathbf{x})} \left[ score(\hat{\mathbf{Y}}, \check{\mathbf{Y}}) \right] \overset{(a)}{=} \min_{\check{P}(\check{\mathbf{y}}|\mathbf{x}) \in \Xi} \max_{\hat{P}(\hat{\mathbf{y}}|\mathbf{x})} \mathbb{E}_{\tilde{P}(\mathbf{x})\hat{P}(\hat{\mathbf{y}}|\mathbf{x})\check{P}(\check{\mathbf{y}}|\mathbf{x})} \left[ score(\hat{\mathbf{Y}}, \check{\mathbf{Y}}) \right]$$

$$\overset{(b)}{=} \max_{\theta} \left( \mathbb{E}_{\tilde{P}(\mathbf{y},\mathbf{x})} \left[ \theta \cdot \Phi(\mathbf{X}, \mathbf{Y}) \right] + \sum_{\mathbf{x} \in \mathcal{X}} \tilde{P}(\mathbf{x}) \min_{\check{P}(\check{\mathbf{y}}|\mathbf{x})} \max_{\hat{P}(\hat{\mathbf{y}}|\mathbf{x})} \left( \underbrace{score(\hat{\mathbf{y}}, \check{\mathbf{y}}) - \theta \cdot \Phi(\mathbf{x}, \check{\mathbf{y}})}_{C'_{\hat{\mathbf{y}},\check{\mathbf{y}}}} \right) \right), \tag{2}$$

*where: $\Phi(\mathbf{x}, \mathbf{y})$ is a vector of features characterizing the set of prediction variables $\{y_i\}$ and provided contextual variables $\{x_i\}$ each related to predicted variable $y_i$.*

*Proof (sketch).* Equality (a) is a consequence of duality in zero-sum games [29]. Equality (b) is obtained by writing the Lagrangian and taking the dual. Strong Lagrangian duality is guaranteed when a feasible solution exists on the relative interior of the convex constraint set $\Xi$ [2]. (A small amount of slack corresponds to regularization of the $\theta$ parameter in the dual and guarantees the strong duality feasibility requirement is satisfied in practice.) □

The resulting game's payoff matrix can be expressed as the original game scores of Eq. (1) augmented with Lagrangian potentials. The combination defines a new payoff matrix with entries $C'_{\hat{\mathbf{y}},\check{\mathbf{y}}} = score(\hat{\mathbf{y}}, \check{\mathbf{y}}) - \theta \cdot \Phi(\mathbf{x}, \check{\mathbf{y}})$, as shown in Eq. (2).

### 3.2 Example multivariate prediction games and small-scale solutions

Examples of the Lagrangian payoff matrices for the P@2, F-score, and DCG games are shown in Table 1 for three variables. We employ additive feature functions, $\Phi(\mathbf{x}, \check{\mathbf{y}}) = \sum_{i=1}^{n} \phi(x_i) I(\check{y}_i = 1)$,

Table 1: The payoff matrices for the zero-sum games between player $\check{Y}$ choosing columns and player $\hat{Y}$ choosing rows with three variables for: *precision at k* (top); F-score (middle) and DCG with binary relevance values, $\check{y}_i \in \{0,1\}$, and we let $\lg 3 \triangleq \log_2 3$ (bottom).

| P@2 | 000 | 001 | 010 | 011 | 100 | 101 | 110 | 111 |
|---|---|---|---|---|---|---|---|---|
| **011** | $0$ | $\frac{1}{2}-\psi_3$ | $\frac{1}{2}-\psi_2$ | $1-\psi_2-\psi_3$ | $0-\psi_1$ | $\frac{1}{2}-\psi_1-\psi_3$ | $\frac{1}{2}-\psi_1-\psi_2$ | $1-\psi_1-\psi_2-\psi_3$ |
| **101** | $0$ | $\frac{1}{2}-\psi_3$ | $0-\psi_2$ | $\frac{1}{2}-\psi_2-\psi_3$ | $\frac{1}{2}-\psi_1$ | $1-\psi_1-\psi_3$ | $\frac{1}{2}-\psi_1-\psi_2$ | $1-\psi_1-\psi_2-\psi_3$ |
| **110** | $0$ | $0-\psi_3$ | $\frac{1}{2}-\psi_2$ | $\frac{1}{2}-\psi_2-\psi_3$ | $\frac{1}{2}-\psi_1$ | $\frac{1}{2}-\psi_1-\psi_3$ | $1-\psi_1-\psi_2$ | $1-\psi_1-\psi_2-\psi_3$ |

| $F_1$ | 000 | 001 | 010 | 011 | 100 | 101 | 110 | 111 |
|---|---|---|---|---|---|---|---|---|
| **000** | $1$ | $0-\psi_3$ | $0-\psi_2$ | $0-\psi_2-\psi_3$ | $0-\psi_1$ | $0-\psi_1-\psi_3$ | $0-\psi_1-\psi_2$ | $0-\psi_1-\psi_2-\psi_3$ |
| **001** | $0$ | $1-\psi_3$ | $0-\psi_2$ | $\frac{2}{3}-\psi_2-\psi_3$ | $0-\psi_1$ | $\frac{2}{3}-\psi_1-\psi_3$ | $0-\psi_1-\psi_2$ | $\frac{1}{2}-\psi_1-\psi_2-\psi_3$ |
| **010** | $0$ | $0-\psi_3$ | $1-\psi_2$ | $\frac{2}{3}-\psi_2-\psi_3$ | $0-\psi_1$ | $0-\psi_1-\psi_3$ | $\frac{2}{3}-\psi_1-\psi_2$ | $\frac{1}{2}-\psi_1-\psi_2-\psi_3$ |
| **011** | $0$ | $\frac{2}{3}-\psi_3$ | $\frac{2}{3}-\psi_2$ | $1-\psi_2-\psi_3$ | $0-\psi_1$ | $\frac{1}{2}-\psi_1-\psi_3$ | $\frac{1}{2}-\psi_1-\psi_2$ | $\frac{4}{5}-\psi_1-\psi_2-\psi_3$ |
| **100** | $0$ | $0-\psi_3$ | $0-\psi_2$ | $0-\psi_2-\psi_3$ | $1-\psi_1$ | $\frac{2}{3}-\psi_1-\psi_3$ | $\frac{2}{3}-\psi_1-\psi_2$ | $\frac{1}{2}-\psi_1-\psi_2-\psi_3$ |
| **101** | $0$ | $\frac{2}{3}-\psi_3$ | $0-\psi_2$ | $\frac{1}{2}-\psi_2-\psi_3$ | $\frac{2}{3}-\psi_1$ | $1-\psi_1-\psi_3$ | $\frac{1}{2}-\psi_1-\psi_2$ | $\frac{4}{5}-\psi_1-\psi_2-\psi_3$ |
| **110** | $0$ | $0-\psi_3$ | $\frac{2}{3}-\psi_2$ | $\frac{1}{2}-\psi_2-\psi_3$ | $\frac{2}{3}-\psi_1$ | $\frac{1}{2}-\psi_1-\psi_3$ | $1-\psi_1-\psi_2$ | $\frac{4}{5}-\psi_1-\psi_2-\psi_3$ |
| **111** | $0$ | $\frac{1}{2}-\psi_3$ | $\frac{1}{2}-\psi_2$ | $\frac{4}{5}-\psi_2-\psi_3$ | $\frac{1}{2}-\psi_1$ | $\frac{4}{5}-\psi_1-\psi_3$ | $\frac{4}{5}-\psi_1-\psi_2$ | $1-\psi_1-\psi_2-\psi_3$ |

| DCG | 000 | 001 | 010 | 011 | 100 | 101 | 110 | 111 |
|---|---|---|---|---|---|---|---|---|
| **123** | $0$ | $\frac{1}{2}-\psi_3$ | $\frac{1}{\lg 3}-\psi_2$ | $\frac{1}{2}+\frac{1}{\lg 3}-\psi_2-\psi_3$ | $1-\psi_1$ | $\frac{3}{2}-\psi_1-\psi_3$ | $1+\frac{1}{\lg 3}-\psi_1-\psi_2$ | $\frac{3}{2}+\frac{1}{\lg 3}-\psi_1-\psi_2-\psi_3$ |
| **132** | $0$ | $\frac{1}{\lg 3}-\psi_3$ | $\frac{1}{2}-\psi_2$ | $\frac{1}{2}+\frac{1}{\lg 3}-\psi_2-\psi_3$ | $1-\psi_1$ | $1+\frac{1}{\lg 3}-\psi_1-\psi_3$ | $\frac{3}{2}-\psi_1-\psi_2$ | $\frac{3}{2}+\frac{1}{\lg 3}-\psi_1-\psi_2-\psi_3$ |
| **213** | $0$ | $\frac{1}{2}-\psi_3$ | $1-\psi_2$ | $\frac{3}{2}-\psi_2-\psi_3$ | $\frac{1}{\lg 3}-\psi_1$ | $\frac{1}{2}+\frac{1}{\lg 3}-\psi_1-\psi_3$ | $1+\frac{1}{\lg 3}-\psi_1-\psi_2$ | $\frac{3}{2}+\frac{1}{\lg 3}-\psi_1-\psi_2-\psi_3$ |
| **231** | $0$ | $\frac{1}{\lg 3}-\psi_3$ | $1-\psi_2$ | $1+\frac{1}{\lg 3}-\psi_2-\psi_3$ | $\frac{1}{2}-\psi_1$ | $\frac{1}{2}+\frac{1}{\lg 3}-\psi_1-\psi_3$ | $\frac{3}{2}-\psi_1-\psi_2$ | $\frac{3}{2}+\frac{1}{\lg 3}-\psi_1-\psi_2-\psi_3$ |
| **312** | $0$ | $1-\psi_3$ | $\frac{1}{2}-\psi_2$ | $\frac{3}{2}-\psi_2-\psi_3$ | $\frac{1}{\lg 3}-\psi_1$ | $1+\frac{1}{\lg 3}-\psi_1-\psi_3$ | $\frac{1}{2}+\frac{1}{\lg 3}-\psi_1-\psi_2$ | $\frac{3}{2}+\frac{1}{\lg 3}-\psi_1-\psi_2-\psi_3$ |
| **321** | $0$ | $1-\psi_3$ | $\frac{1}{\lg 3}-\psi_2$ | $1+\frac{1}{\lg 3}-\psi_2-\psi_3$ | $\frac{1}{2}-\psi_1$ | $\frac{3}{2}-\psi_1-\psi_3$ | $\frac{1}{2}+\frac{1}{\lg 3}-\psi_1-\psi_2$ | $\frac{3}{2}+\frac{1}{\lg 3}-\psi_1-\psi_2-\psi_3$ |

in these examples (with indicator function $I(\cdot)$). We compactly represent the Lagrangian potential terms for each game with potential variables, $\psi_i \triangleq \theta \cdot \phi(X_i = x_i)$ when $\check{Y}_i = 1$ (and 0 otherwise).

Zero-sum games such as these can be solved using a pair of linear programs that have a constraint for each pure action (set of variable assignments) in the game [29]:

$$\max_{v,\hat{P}(\hat{\mathbf{y}}|\mathbf{x})\geq 0} v \text{ such that } v \leq \sum_{\hat{\mathbf{y}}\in\mathcal{Y}} \hat{P}(\hat{\mathbf{y}}|\mathbf{x})\boldsymbol{C}'_{\hat{\mathbf{y}},\check{\mathbf{y}}} \ \forall \hat{\mathbf{y}}\in\mathcal{Y} \text{ and } \sum_{\hat{\mathbf{y}}\in\mathcal{Y}} \hat{P}(\hat{\mathbf{y}}|\mathbf{x}) = 1; \qquad (3)$$

$$\min_{v,\check{P}(\check{\mathbf{y}}|\mathbf{x})\geq 0} v \text{ such that } v \geq \sum_{\check{\mathbf{y}}\in\mathcal{Y}} \check{P}(\check{\mathbf{y}}|\mathbf{x})\boldsymbol{C}'_{\hat{\mathbf{y}},\check{\mathbf{y}}} \ \forall \check{\mathbf{y}}\in\mathcal{Y} \text{ and } \sum_{\check{\mathbf{y}}\in\mathcal{Y}} \check{P}(\check{\mathbf{y}}|\mathbf{x}) = 1, \qquad (4)$$

where $\boldsymbol{C}'$ is the Lagrangian-augmented payoff and $v$ is the value of the game. The second player to act in a zero-sum game can maximize/minimize using a pure strategy (i.e., a single value assignment to all variables). Thus, these LPs consider only the set of pure strategies of the opponent to find the first player's mixed equilibrium strategy. The equilibrium strategy for the predictor is a distribution over rows and the equilibrium strategy for the adversary is a distribution over columns.

The size of each game's payoff matrix grows exponentially with the number of variables, $n$: $(2^n)\binom{n}{k}$ for the *precision at k* game; $(2^n)^2$ for the F-score game; and $(n!\ k^n)$ for the DCG game with $k$ possible relevance levels. These sizes make explicit construction of the game matrix impractical for all but the smallest of problems.

### 3.3 Large-scale strategy inference

More efficient methods for obtaining Nash equilibria are needed to scale our MPG approach to large prediction tasks with exponentially-sized payoff matrices. Though much attention has focused on efficiently computing $\epsilon$-Nash equilibria (e.g., in $\mathcal{O}(1/\epsilon)$ time or $\mathcal{O}(\ln(1/\epsilon))$ time [8]), which guarantee each player a payoff within $\epsilon$ of optimal, we employ an approach for finding an exact equilibrium that works well in practice despite not having as strong theoretical guarantees [20].

Consider the reduced game matrices of Table 2. The Nash equilibrium for the *precision at k* game with Lagrangian potentials $\psi_1 = \psi_2 = \psi_3 = 0.4$ is: $\hat{P}(\hat{\mathbf{y}}|\mathbf{x}) = \begin{bmatrix} \frac{1}{3} & \frac{1}{3} & \frac{1}{3} \end{bmatrix}$ and $\check{P}(\check{\mathbf{y}}|\mathbf{x}) = \begin{bmatrix} \frac{1}{3} & \frac{1}{3} & \frac{1}{3} \end{bmatrix}$; with a game value of $-\frac{2}{15}$. The Nash equilibrium for the reduced F-score game with no learning (i.e., $\psi_1 = \psi_2 = \psi_3 = 0$) is: $\hat{P}(\hat{\mathbf{y}}|\mathbf{x}) = \begin{bmatrix} \frac{1}{3} & \frac{2}{3} \end{bmatrix}$ and $\check{P}(\check{\mathbf{y}}|\mathbf{x}) = \begin{bmatrix} \frac{1}{3} & \frac{2}{9} & \frac{2}{9} & \frac{2}{9} \end{bmatrix}$; with a game value of $\frac{2}{3}$. The reduced game equilibrium is also an equilibrium of the original game. Though the exact size of the subgame and its specific actions depends on the values of $\psi$, often a compact sub-game with identical equilibrium or close approximation exists [18]. Motivated by the compactness of the reduced game, we employ a constraint generation approach known as the double oracle algorithm [20] to iteratively construct an appropriate reduced game that provides the correct equilibrium but avoids the computational complexity of the original exponentially sized game.

Table 2: The reduced *precision at k* game with $\psi_1 = \psi_2 = \psi_3 = 0.4$ (top) and F-score game with $\psi_1 = \psi_2 = \psi_3 = 0$ (bottom).

|     | 011 | 101 | 110 |
|-----|-----|-----|-----|
| **011** | 0.2 | -0.3 | -0.3 |
| **101** | -0.3 | 0.2 | -0.3 |
| **111** | -0.3 | -0.3 | 0.2 |

|     | 000 | 001 | 010 | 100 |
|-----|-----|-----|-----|-----|
| **000** | 0 | 1 | 1 | 1 |
| **111** | 1 | $\frac{1}{2}$ | $\frac{1}{2}$ | $\frac{1}{2}$ |

---

**Algorithm 1** Constraint generation game solver

---

**Input:** Lagrange potentials for each variable, $\boldsymbol{\psi} = \{\psi_1, \psi_2, \ldots, \psi_n\}$; initial action sets $\hat{S}_0$ and $\check{S}_0$
**Output:** Nash equilibrium, $\left( \hat{P}(\hat{\mathbf{y}}|\mathbf{x}), \check{P}(\check{\mathbf{y}}|\mathbf{x}) \right)$

1: Initialize Player $\hat{Y}$'s action set $\hat{S} \leftarrow \hat{S}_0$ and Player $\check{Y}$'s action set $\check{S} \leftarrow \check{S}_0$
2: $\boldsymbol{C}' \leftarrow$ buildPayoffMatrix($\hat{S}, \check{S}, \psi$)  $\qquad \triangleright$ Using Eq. (2) for the sub-game matrix of $\hat{S} \times \check{S}$
3: **repeat**
4: $\quad [\hat{P}(\hat{\mathbf{y}}|\mathbf{x}), v_{\mathrm{Nash}_1}] \leftarrow$ solveZeroSumGame$_{\check{Y}}(\boldsymbol{C}')$ $\qquad\qquad \triangleright$ Using the LP of Eq. (3)
5: $\quad [\check{a}, \check{v}_{\mathrm{BR}}] \leftarrow$ findBestResponseAction($P(\hat{\mathbf{y}}|\mathbf{x}), \psi$) $\quad \triangleright \check{a}$ denotes the best response action
6: $\quad$ **if** ($v_{\mathrm{Nash}_1} \neq \check{v}_{\mathrm{BR}}$) **then** $\qquad\qquad \triangleright$ Check if best response provides improvement
7: $\qquad \check{S} \leftarrow \check{S} \cup \check{a}$
8: $\qquad \boldsymbol{C}' \leftarrow$ buildPayoffMatrix($\hat{S}, \check{S}, \psi$) $\qquad\qquad \triangleright$ Add new row to game matrix
9: $\quad$ **end if**
10: $\quad [\check{P}(\check{\mathbf{y}}|\mathbf{x}), v_{\mathrm{Nash}_2}] \leftarrow$ solveZeroSumGame$_{\hat{Y}}(\boldsymbol{C}')$ $\qquad\qquad \triangleright$ Using the LP of Eq. (4)
11: $\quad [\hat{a}, \hat{v}_{\mathrm{BR}}] \leftarrow$ findBestResponseAction($P(\check{\mathbf{y}}|\mathbf{x}), \psi$)
12: $\quad$ **if** ($v_{\mathrm{Nash}_2} \neq \hat{v}_{\mathrm{BR}}$) **then**
13: $\qquad \hat{S} \leftarrow \hat{S} \cup \hat{a}$
14: $\qquad \boldsymbol{C}' \leftarrow$ buildPayoffMatrix($\hat{S}, \check{S}, \psi$) $\qquad\qquad \triangleright$ Add new column to game matrix
15: $\quad$ **end if**
16: **until** ($v_{\mathrm{Nash}_1} = v_{\mathrm{Nash}_2} = \hat{v}_{\mathrm{BR}} = \check{v}_{\mathrm{BR}}$) $\qquad \triangleright$ Stop if neither best response provides improvement
17: **return** $[\hat{P}(\hat{\mathbf{y}}|\mathbf{x}), \check{P}(\check{\mathbf{y}}|\mathbf{x})]$

---

Neither player can improve upon their strategy with additional pure
strategies when Algorithm 1 terminates, thus the mixed strategies it returns are a Nash equilibrium pair [20]. Additionally, the algorithm is efficient in practice so long as each player's strategy is compact (i.e., the number of actions with non-zero probability is a polynomial subset of the label combinations) and best responses to opponents' strategies can be obtained efficiently (i.e., in polynomial time) for each player. Additionally, this algorithm can be modified to find approximate equilibria by limiting the number of actions for each player's set $\hat{\mathcal{S}}$ and $\check{\mathcal{S}}$.

### 3.4 Efficiently computing best responses

The tractability of our approach largely rests on our ability to efficiently find best responses to opponent strategies: $\mathrm{argmax}_{\hat{y} \in \hat{\mathcal{Y}}} \mathbb{E}_{\check{P}(\check{y}|x)}[\boldsymbol{C}'_{\hat{y}, \check{Y}}]$ and $\mathrm{argmin}_{\check{y} \in \check{\mathcal{Y}}} \mathbb{E}_{\hat{P}(\hat{y}|x)}[\boldsymbol{C}'_{\hat{Y}, \check{y}}]$. For some combinations of loss functions and features, finding the best response is trivial using, e.g., a greedy selection algorithm. Other loss function/feature combinations require specialized algorithms or are NP-hard. We illustrate each situation.

**Precision at k best response** Many best responses can be obtained using greedy algorithms that are based on marginal probabilities of the opponent's strategy. For example, the expected payoff in

the *precision at k* game for the estimator player setting $\hat{y}_i = 1$ is $\check{P}(\check{y}_i = 1|x)$. Thus, the set of top $k$ variables with the largest marginal label probability provides the best response. For the adversary's best response, the Lagrangian terms must also be included. Since $k$ is a known variable, as long as the value of each included term, $\hat{P}(\hat{y}_i = 1, ||\hat{y}||_1 = k|x) - k\psi_i$, is negative, the sum is the smallest, and the corresponding response is the best for the adversary.

**F-score game best response** We leverage a recently developed method for efficiently maximizing the F-score when a distribution over relevant documents is given [6]. The key insight is that the problem can be separated into an inner greedy maximization over item sets of a certain size $k$ and an outer maximization to select the best set size $k$ from $\{0, \ldots, n\}$. This method can be directly applied to find the best response of the estimator player, $\hat{Y}$, since the Lagrangian terms of the cost matrix are invariant to the choice of $\hat{y}$. Algorithm 2 obtains the best response for the adversary player, $\check{Y}$, using slight modifications to incorporate the Lagrangian potentials into the objective function.

---

**Algorithm 2** Lagrangian-augmented F-measure Maximizer for adversary player $\check{Y}$

---

**Input:** vector $\hat{P}$ of estimator probabilities and Lagrange potentials $\psi$ $(\psi_1, \psi_2, ..., \psi_n)$
1: define matrix $\boldsymbol{W}$ with element $\boldsymbol{W}_{s,k} = \frac{1}{s+k}, \qquad s, k \in \{1, ..., n\}$
2: construct matrix $\boldsymbol{F} = \hat{P} \times \boldsymbol{W} - \frac{1}{2}\psi^{\mathrm{T}} \times \mathbf{1}^n$          $\triangleright$ $\mathbf{1}^n$ is the all ones $1 \times n$ vector
3: **for** $k = 1$ to $n$ **do**
4:      solve the inner optimization problem:
5:      $\mathbf{a}^{(k)^*} = \text{argmin}_{a \in A_k} 2\sum_{i=1}^{n} a_i f_{ik}$          $\triangleright$ $A_k = \{\mathbf{a} \in \{0,1\}^n | \sum_{i=1}^{n} a_i = k\}$
6:      by setting $a_i^{(k)} = 1$ for the $k$-th column of $\boldsymbol{F}$'s **smallest** $k$ elements, and $a_i = 0$ for the rest;
7:      store a value of $\mathbb{E}_{\mathbf{y} \sim p(\hat{Y}|\mathbf{x})}[\boldsymbol{F}(\mathbf{y}, \mathbf{a}^{(k)^*})] = 2\sum_{i=1}^{n} a_i^{(k)^*} f_{ik}$
8: **end for**
9: for $k = 0$ take $\mathbf{a}^{(k)^*} = \mathbf{0^n}$, and $\mathbb{E}_{\mathbf{y} \sim P(\hat{Y}|\mathbf{x})}[\boldsymbol{F}(\mathbf{y}, \mathbf{0^n})] = p(\hat{Y} = \mathbf{0^n}|\mathbf{x})$
10: solve the outer optimization problem:
11: $\mathbf{a}^* = \text{argmin}_{\mathbf{a} \in \{\mathbf{a}^{(0)^*}, ..., \mathbf{a}^{(n)^*}\}} \mathbb{E}_{\mathbf{y} \sim p(\hat{Y}|\mathbf{x})}[\boldsymbol{F}(\mathbf{y}, \mathbf{a})]$
12: **return** $\mathbf{a}^*$ and $\mathbb{E}_{\mathbf{y} \sim p(\hat{Y}|\mathbf{x})}[\boldsymbol{F}(\mathbf{y}, \mathbf{a}^*)]$

---

**Order inversion best response** Another common loss measure when comparing two rankings is the number of pairs of items with inverted order across rankings (i.e., one variable may occur before another in one ranking, but not in the other ranking). Only the marginal probabilities of pairwise orderings, $\hat{P}(\hat{\sigma}^{-1}(i) > \hat{\sigma}^{-1}(j)) \triangleq \sum_{\hat{\sigma}} \hat{P}(\hat{\sigma}) \ I(\sigma^{-1}(i) > \sigma^{-1}(j))$, are needed to construct the portion of the payoff received for $\check{\sigma}$ ranking item $i$ over item $j$, $\hat{P}(\hat{\sigma}^{-1}(i) > \hat{\sigma}^{-1}(j))(1 + \psi_{i>j})$, where $\psi_{i>j}$ is a Lagrangian potential based on pair-wise features for ranking item $i$ over item $j$. One could construct a fully connected directed graph with edges weighted by these portions of the payoff for ranking pairs of items. The best response for $\check{\sigma}$ corresponds to a set of acyclic edges with the smallest sum of edge weights. Unfortunately, this problem is NP-hard in general because the NP-complete minimum feedback arc set problem [15], which seeks to form an acyclic graph by removing the set of edges with the minimal sum of edge weights, can be reduced to it.

**DCG best response** Although we cannot find an efficient algorithm to get the best response using order inversion, solving best response of DCG has a known efficient algorithm. In this problem the maximizer is a permutation of the documents while the minimizer is the relevance score of each document pair. The estimator's best response $\hat{\sigma}$ maximizes:

$$\sum_{\check{y}} P(\check{y}|x) \left( \sum_{i=1}^{n} \frac{2^{\check{y}_{\hat{\sigma}(i)}} - 1}{\log_2(i+1)} - \theta \cdot \phi(x, \check{y}) \right) = \sum_{i=1}^{n} \frac{1}{\log_2(i+1)} \left( \sum_{\check{y}} P(\check{y}|x) 2^{\check{y}_{\hat{\sigma}(i)}} - 1 \right) - c,$$

where $c$ is a constant that has no relationship with $\hat{\sigma}$. Since $1/\log_2(i+1)$ is monotonically decreasing, computing and sorting $\sum_{\check{y}} P(\check{y}|x) 2^{\check{y}_i} - 1$ with descending order and greedily assign the order to $\hat{\sigma}$ is optimal. The adversary's best response using additive features minimizes:

$$\sum_{\hat{\sigma}} P(\hat{\sigma}|x) \sum_{i=1}^{n} \frac{2^{\check{y}_{\hat{\sigma}(i)}} - 1}{\log_2(i+1)} - \sum_{i=1}^{n} \theta_i \cdot \phi_i(x_i, \check{y}_i) = \sum_{i=1}^{n} \left( \sum_{\hat{\sigma}} P(\hat{\sigma}|x) \frac{2^{\check{y}_i} - 1}{\log_2(\sigma^{-1}(i) + 1)} - \theta_i \cdot \phi_i(x_i, \check{y}_i) \right).$$

Thus, by using the expectation of a function of each variable's rank, $1/(\log_2(\sigma^{-1}(i)+1)$, which is easily computed from $\hat{P}(\sigma)$, each variable's relevancy score $\check{y}_i$ can be independently chosen.

## 3.5 Parameter estimation

Predictive model parameters, $\theta$, must be chosen to ensure that the adversarial distribution is similar to training data. Though adversarial prediction can be posed as a convex optimization problem [1], the objective function is not smooth. General subgradient methods require $\mathcal{O}(1/\epsilon^2)$ iterations to provide an $\epsilon$ approximation to the optima. We instead employ L-BFGS [19], which has been empirically shown to converge at a faster rate in many cases despite lacking theoretical guarantees for non-smooth objectives [16]. We also employ $L_2$ regularization to avoid overfitting to the training data sample. The addition of the smooth regularizer often helps to improve the rate of convergence.

The gradient in these optimizations with $L_2$ regularization, $-\frac{\lambda}{2}||\theta||^2$, for training dataset $\mathcal{D} = \{(\mathbf{x}^{(i)}, \mathbf{y}^{(i)})\}$ is the difference between feature moments with additional regularization term: $\frac{1}{|\mathcal{D}|}\sum_{j=1}^{|\mathcal{D}|}\left(\Phi(\mathbf{x}^{(i)}, \mathbf{y}^{(i)}) - \sum_{\check{\mathbf{y}} \in \mathcal{Y}}\check{P}(\check{\mathbf{y}}|\mathbf{x}^{(i)})\Phi(\mathbf{x}^{(i)}, \check{\mathbf{y}})\right) - \lambda\theta$. The adversarial strategies $\check{P}(\cdot|\mathbf{x}^{(i)})$ needed for calculating this gradient are computed via Alg. 1.

# 4 Experiments

We evaluate our approach, Multivariate Prediction Games (MPG), on the three performance measures of interest in this work: *precision at k*, F-score, and DCG. Our primary point of comparison is with structured support vector machines (SSVM)[27] to better understand the trade-offs between convexly approximating the loss function with the hinge loss versus adversarially approximating the training data using our approach. We employ an optical recognition of handwritten digits (OPTDIGITS) dataset [17] (10 classes, 64 features, 3,823 training examples, 1,797 test examples), an income prediction dataset ('a4a' ADULT[1] [17] (two classes, 123 features, 3,185 training examples, 29,376 test examples), and query-document pairs from the million query TREC 2007 (MQ2007) dataset of LETOR4.0 [23] (1700 queries, 41.15 documents on average per query, 46 features per document). Following the same evaluation method used in [27] for OPTDIGITS, the multi-class dataset is converted into multiple binary datasets and we report the macro-average of the performance of all classes on test data. For OPTDIGITS/ADULT, we use a random $\frac{1}{3}$ of the training data as a holdout validation data to select the $L_2$ regularization parameter trade-off $C \in \{2^{-6}, 2^{-5}, ..., 2^6\}$.

We evaluate the performance of our approach and comparison methods (SSVM variants[2] and logistic regression (LR)) using *precision at k*, where $k$ is half the number of positive examples (i.e. $k = \frac{1}{2}POS$), and F-score. For precision at $k$, we restrict the pure strategies of the adversary to select $k$ positive labels. This prevents adversary strategies with no positive labels. From the results in Table 3, we see that our approach, MPG, works better than SSVM on the OPTDGITS datasets: slightly better on *precision at k* and more significantly better on F-measure. For the ADULT dataset, MPG provides equivalent performance for *precision at k* and better performance on F-

Table 3: *Precision at k* (top) and F-score performance (bottom).

| Precision@k | OPTDIGITS | ADULT |
|---|---|---|
| **MPG** | 0.990 | 0.805 |
| **SSVM** | 0.956 | 0.638 |
| **SSVM'** | 0.989 | 0.805 |

| F-score | OPTDIGITS | ADULT |
|---|---|---|
| **MPG** | 0.920 | 0.697 |
| **SSVM** | 0.915 | 0.673 |
| **LR** | 0.914 | 0.639 |

measure. The nature of the running time required for validation and testing is very different for SSVM, which must find the maximizing set of variable assignments, and MPG, which must interactively construct a game and its equilibrium. Model validation and testing require $\approx 30$ seconds for SSVM on the OPTDIGITS dataset and $\approx 3$ seconds on the ADULT dataset, while requiring $\approx 9$ seconds and $\approx 25$ seconds for MPG precision at $k$ and $\approx 1397$ seconds and $\approx 252$ seconds for MPG F-measure optimization, respectively. For precision at $k$, MPG is within an order of magni-

tude (better for OPTDIGITS, worse for ADULT). For the more difficult problem of maximizing the F-score of ADULT over $29,376$ test examples, the MPG game becomes quite large and requires significantly more computational time. Though our MPG method is not as finely optimized as existing SSVM implementations, this difference in run times will remain as the game formulation is inherently more computationally demanding for difficult prediction tasks.

We compare the performance of our approach and comparison methods using five-fold cross validation on the MQ2007 dataset. We measure performance using Normalized DCG (NDCG), which divides the realized DCG by the maximum possible DCG for the dataset, based on a slightly different variant of DCG employed by LETOR4.0: $\text{DCG}''(\hat{\sigma}, \mathbf{y}) = 2^{y_{\hat{\sigma}(1)}} - 1 + \sum_{i=2}^{n} \frac{2^{y_{\hat{\sigma}(i)}} - 1}{\log_2 i}$. The comparison methods are: RankSVM-Struct [13], part of SVM$^{\text{struct}}$ which uses structured SVM to predict the rank; ListNet [3], a list-wise ranking algorithm employing cross entropy loss; AdaRank-NDCG [30], a boosting method using 'weak rankers' and data reweighing to achieve good NDCG performance; AdaRank-MAP uses Mean Average Precision (MAP) rather than NDCG; and RankBoost [7], which reduces ranking to binary classification problems on instance pairs.

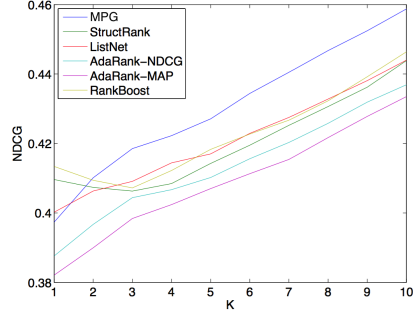

Figure 1: NDCG@K as K increases.

Table 4: MQ2007 NDCG Results.

| Method | Mean NDCG |
|---|---|
| MPG | 0.5220 |
| RankSVM | 0.4966 |
| ListNet | 0.4988 |
| AdaRank-NDCG | 0.4914 |
| AdaRank-MAP | 0.4891 |
| RankBoost | 0.5003 |

Table 4 reports the NDCG@K averaged over all values of $K$ (between 1 and, on average 41) while Figure 1 reports the results for each value of $K$ between 1 and 10. From this, we can see that our MPG approach provides better rankings on average than the baseline methods except when $K$ is very small ($K = 1, 2$). In other words, the adversary focuses most of its effort in reducing the score received from the first item in the ranking, but at the expense of providing a better overall NDCG score for the ranking as a whole.

## 5 Discussion

We have extended adversarial prediction games [1] to settings with multivariate performance measures in this paper. We believe that this is an important step in demonstrating the benefits of this approach in settings where structured support vector machines [14] are widely employed. Our future work will investigate improving the computational efficiency of adversarial methods and also incorporating structured statistical relationships amongst variables in the constraint set in addition to multivariate performance measures.

## Acknowledgments

This material is based upon work supported by the National Science Foundation under Grant No. #1526379, Robust Optimization of Loss Functions with Application to Active Learning.

## Footnotes

[1] http://www.csie.ntu.edu.tw/~cjlin/libsvmtools/datasets/binary.html)

[2]For precision at $k$, the original SSVM's implementation uses the restriction $k$ during training, but not during testing. We modified the code by ordering SSVM's prediction value for each test example, and select the top k predictions as positives, the rest are considered as negatives. We denote the original implementation as SSVM, and the modified version as SSVM'.

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
