[Reviews · NeurIPS 2015]

Submitted by Assigned_Reviewer_1

The approach presented in this paper - adversarial learning under multivariate performance measures - seems very interesting and original. The presentation is very clear and appropriately gradual also for a reader like myself who was new to the framework. The method is intriguing in its own right; in addition, the experimental results show that, with appropriate design choices, it can outperform structural SVM.

In my opinion, the paper could be published as is. I just add some comments that should help its readability for a broader audience:

1. It is true that the paper addresses a "structured prediction" problem. This is because the use of multivariate performance measures "couples" the samples even if they are i.i.d.. However, it seems to me that in the literature the term "structured prediction" is universally understood as the structuring of the output labels, i.e., feature functions which factorise over graphs, trees and the like. The paper does not address this case. In fact, the 2005 paper from Thorsten Joachims does not use "structured" or "structural" in the title. I therefore (warmly) suggest changing the title of this paper to something like "Adversarial learning under multivariate losses". The possible structuring of the feature functions can be mentioned in the body.

2. On a similar rationale, it took me a bit to map the word "score" as used in the paper to the "performance measure" (although it is clearly stated at lines 062-063). It is just not very intuitive since most people associate "score" with \theta * \phi(x,y). Why not change word "score" to "perf"? Then, loss becomes 1 - perf.

3. Table 1 is really a boon since it allowed me to grasp the technique. It would help to say explicitly that, here, \check{y} is the prediction and \hat{y} acts as the ground truth.

4. I find the justification for eq. (7) too brief. Please add some passages to explain how the l.h.s. in the constraint becomes like this. In addition: I think \tilde{P}(x) should be removed.

5. In section 3.5, before the gradient in \theta, please write the complete objective with a brief explanation.

Only typo found: p.6, line 362: "demonstate"
Summary: This paper presents a learning framework for multivariate performance measures (e.g., precision at k, F1 score etc) based on an "adversarial formulation for classification" (i.e., a two-player game aiming at an equilibrium distribution). The primary reference is a paper recently published at UAI 2015 (i.e., reference [1]). The main contribution of this follow-up paper is a substantial extension for various multivariate performance measures for prediction and ranking.

The main term of comparison for the proposed approach is maximum-margin/cutting plane training for the same performance measures. A range of experimental results show that the proposed technique is competitive and can outperform SSVM in a number of cases.

I believe this paper is very interesting, well written and very original and I definitely recommend it for acceptance.

Submitted by Assigned_Reviewer_2

line 95: "losses are often not convex" - the losses considered here are functions over a discrete domain - convexity is probably not the right word here.
Summary: The paper proposes a game theoretic approach to multi-label prediction with non-decomposable losses.

The proposed approach is an interesting alternative to a structured output SVM, and analysis is given for P@2, F-score, and DCG losses.

Theoretical results are interesting, while empirical results are somewhat underwhelming, with no error bars, and limited comparison on a couple datasets.

Submitted by Assigned_Reviewer_3

PAC-Bayes: please take a look at McAllester & Keshet 2011 and Keshet et al. 2010.

Summary: The paper is interesting but might be a little pre-mature for publication in ML venue. There are other frameworks that fit

structured outputs (e.g., PAC-Bayes) and do not require convex losses (although the risk is still non-convex). Also, what might be the generalization properties of such games?

Submitted by Assigned_Reviewer_4

Summary: Learning structured predictors using two player zero sum games is discussed within this work. Instead of employing a convex approximation to the task loss as traditionally done, the exact loss is minimized using `approximations of the data,' i.e., distributions. A two player game is formulated which is intractable in general. Approximations similar to column generation techniques are employed and shown to work well in practice.

Comments: - Theorem 1 is the main work horse employed within the submission. As a reader I'd therefore appreciate a more detailed sketch of the proof. Particularly, referring to step (a) as duality seems counter-intuitive to me at the moment. Does this step require the score function to be symmetric? Moreover I'm wondering whether step (b) is just the Lagrangian. Why do the authors argue that `taking the dual' (l.161) is required which would correspond to carrying out the maximization or minimization?

- A technique very related to the proposed approach is direct loss minimization as discussed in work by McAllester et al. (`Direct Loss Minimization for Structured Prediction', NIPS 2010). I'm missing a discussion and possibly a comparison.

- Besides accuracy and theoretical convergence guarantees, a wall clock time comparison for training predictors is desirable for a reader. Could the authors comment?
Summary: Learning structured predictors using zero sum games is an interesting new idea. The employed Theorem should be discussed more rigorously and some related work could be discussed more carefully.

Submitted by Assigned_Reviewer_5

This paper presents to formulate multivariate prediction as a game between a loss minimizer and a loss-maximizing evaluation player. The authors also discuss efficient learning algorithm that adding actions during the training to avoid the exponential number of rows in the game matrix. The idea is interesting and well-motivated.

Quality and Clarity: the paper is well-written and provide several nice examples to help readers understand the concepts.

Originality: Although there are a few paper discussed about using structured SVM with multivariate losses, I'm not aware

of other approaches that form this problem as an adversarial game.

Significance: Learning with multivariate losses is an important problem, and the paper presents another direction for solving this problem.

Some comments/questions:

- The derivation of second equality (b) in Equation (7) is still unclear to me.

- Is there a good reason why SMP performs badly on ADULT when optimizing precision@k but performs well when optimizing F-score?

- I wonder if the authors can provide the training time of SMP and SSVM.
Summary: This paper present a new approach based on game theory for learning with multivariate losses. The paper is well-written and provide several nice examples. The experiments are pretty comprehensive and demonstrate that the proposed algorithm is competitive to the structured learning model.

Author Feedback
Author rebuttal: We thank the reviewers for their useful comments!

Review #1
We will modify our paper as suggested, changing the title to "Adversarial Prediction Games for Multivariate Losses."

Review #2
- The derivation of second equality (b) in Equation (7) is still unclear to me.
>>>> We will add a more detailed proof to our revision. See also our response to Review #4.

- Is there a good reason why SMP performs badly on ADULT when optimizing precision@k but performs well when optimizing F-score?
>>>> There seem to be two related reasons. First, the precision@k metric tends to provide a great deal of flexibility to the adversary; the predictor is restricted to making k < POS positive label predictions, allowing the adversary to place more probability on the remaining POS examples not chosen. F-score does not provide such flexibility. Second, the SSVM implementation incorporates constraints for all values of k whereas SMP considers games where the predictor chooses exactly k positive labels (see also footnote 3). We investigate an equal footing (e.g., training SMP using all choices of k, or SSVM with fixed k) in our revision.

- I wonder if the authors can provide the training time of SMP and SSVM.
>>>> SMP is roughly 5~10 times slower than SSVM depending on the complexity of finding the best response during each iteration.

Review #3 (missing)

Review #4
- Theorem 1 .... referring to step (a) as duality seems counter-intuitive to me at the moment. Does this step require the score function to be symmetric? .... Why do the authors argue that `taking the dual' (l.161) is required which would correspond to carrying out the maximization or minimization?
>>>> Our revision elaborates on this proof and distinguishes between duality for zero-sum games and Lagrangian duality (both are used). In short, going from the Lagrangian primal, min_{\check{P}} max_\theta L(.,.), to the dual, max_\theta min_{\check{P}} L(.,.), allows \check{P} to be optimized independently within a zero-sum game for each example x.

- A technique very related to the proposed approach is direct loss minimization as discussed in work by McAllester et al. (`Direct Loss Minimization for Structured Prediction', NIPS 2010). I'm missing a discussion and possibly a comparison.
>>>> We will add a discussion of local optimization methods, which likewise seek to more directly optimize the performance measure, but do so in a manner that is susceptible to local optima. See also our response to Review #6.

- Besides accuracy and theoretical convergence guarantees, a wall clock time comparison for training predictors is desirable for a reader. Could the authors comment?
>>>> Please see our response to review #2

Review #5
- There are other frameworks that fit structured outputs (e.g., PAC-Bayes) and do not require convex losses (although the risk is still non-convex).
>>>> To our knowledge, existing methods built on PAC-Bayes bounds employ local optimization (Germain et al. ICML 2009) or sequential Monte Carlo (Ridgway et al. NIPS 2014) -- both susceptible to local optima -- when employed for performance measures similar to those we consider.

- Also, what might be the generalization properties of such games?
>>>> Following [1]: since worst-case assumptions are employed to construct the game, generalization performance is bounded by the expected loss from the minimax game so long as the constraint set \Xi contains the true conditional distribution P(y|x). This can be accomplished with high probability by adding some L_1 or L_2 slack to the constraint set based on finite sample bounds of the measured moments, leading to regularization in the dual.

Review #6
- empirical results are somewhat underwhelming, with no error bars, and limited comparison on a couple datasets.
>>>> These three datasets that have been investigated in related work and enable evaluation of our approach under the three distinct performance measures. NDCG evaluations show consistent improvements over five previous techniques when k > 2. We will add significance analysis in our revision.

- "losses are often not convex" - the losses considered here are functions over a discrete domain - convexity is probably not the right word here.
>>>> We will change this to convey that finding the optimal parametric predictor (e.g., empirical risk minimization) is typically a non-convex optimization problem that is NP-hard for even the simple 0-1 loss.